# Laboratory Automation in Microbiology: Impact on Turnaround Time of Microbiological Samples in COVID Time

**DOI:** 10.3390/diagnostics13132243

**Published:** 2023-07-01

**Authors:** Carla Fontana, Marco Favaro, Marco Pelliccioni, Silvia Minelli, Maria Cristina Bossa, Anna Altieri, Carlo D’Orazi, Federico Paliotta, Oriana Cicchetti, Marilena Minieri, Carla Prezioso, Dolores Limongi, Cartesio D’agostini

**Affiliations:** 1National Institute for Infectious Diseases Lazzaro Spallanzani IRCCS, 00149 Rome, Italy; 2Department of Experimental Medicine, University of Rome Tor Vergata, 00133 Rome, Italy; favaro@uniroma2.it (M.F.); minieri@med.uniroma2.it (M.M.); cartesio.dagostini@ptvonline.it (C.D.); 3Laboratory of Microbiology, Polyclinic of “Tor Vergata”, 00133 Rome, Italy; marco.pelliccioni@uniroma2.it (M.P.); silvia.minelli@ptvonline.it (S.M.); mariacristina.bossa@ptvonline.it (M.C.B.); anna.altieri@ptvonline.it (A.A.); federico.paliotta@ptvonline.it (F.P.); oriana.cicchetti@ptvonline.it (O.C.); 4Department of Laboratory Medicine, Polyclinic Tor Vergata, Viale Oxford 81, 00133 Rome, Italy; 5Laboratory of Microbiology of Chronic-Neurodegenerative Diseases, IRCCS San Raffaele Roma, 00166 Rome, Italy; carla.prezioso@sanraffaele.it (C.P.); dolores.limongi@sanraffaele.it (D.L.); 6Department of Human Sciences and Quality of Life Promotion, San Raffaele University, 00166 Rome, Italy

**Keywords:** fast track, staff shortage, total lab automation, turn-around time, game changer game changer, laboratory workflows, enhance laboratory performance

## Abstract

Background: Laboratory Automation (LA) is an innovative technology that is currently available for microbiology laboratories. LA can be a game changer by revolutionizing laboratory workflows through efficiency improvement and is also effective in the organization and standardization of procedures, enabling staff requalification. It can provide an important return on investment (time spent redefining the workflow as well as direct costs of instrumentation) in the medium to long term. Methods: Here, we present our experience with the WASPLab^®^ system introduced in our lab during the COVID-19 pandemic. We evaluated the impact due to the system by comparing the TAT recorded on our samples before, during, and after LA introduction (from 2019 to 2021). We focused our attention on blood cultures (BCs) and biological fluid samples (BLs). Results: TAT recorded over time showed a significant decrease: from 97 h to 53.5 h (Δ43.5 h) for BCs and from 73 h to 58 h (Δ20 h) for BLs. Despite the introduction of the WASPLab^®^ system, we have not been able to reduce the number of technical personnel units dedicated to the microbiology lab, but WASPLab^®^ has allowed us to direct some of the staff resources toward other laboratory activities, including those required by the pandemic. Conclusions: LA can significantly enhance laboratory performance and, due to the significant reduction in reporting time, can have an effective impact on clinical choices and therefore on patient outcomes. Therefore, the initial costs of LA adoption must be considered worthwhile.

## 1. Introduction

Despite the great progress made in microbiological technology over time, bacteriology remains essentially manual [1]. The first examples of automations were those regarding blood culture processing and identification and sensitivity testing of pathogens, as well as the automated seeding of samples, otherwise known as the Walk Away Specimen Processor [2]. More recently, technological developments, including robotization and imaging (e.g., digital pictures of solid plate culture media), have opened a new chapter in microbiology automation [3]. In bacteriology, daily laboratory work has been exhausting due to the effects of the COVID-19 pandemic as a result of the increase in requests for laboratory tests, staff shortages, and the increased pressure to provide test results in the shortest time possible; thus, it is necessary as never before to invest in automation [3,4,5]. Indeed, microbiological automation offers several advantages: it improves productivity, standardization, traceability, and efficacy and allows a general decrease in turnaround time (TAT) [3,4]. In addition, digital imaging enhances reality, providing microbiologists with an opportunity to see what they were unable to see before and opening a path to tele-bacteriology [1]. Currently, two major systems for lab automation are commercialized: Kiestra^®^ by Becton Dickinson and WASPLab^®^ by Copan [3]. A thorny question in choosing automation is on which criteria to base the choice of the most suitable system for one’s own reality. When microbiologists choose to proceed via lab automation, several aspects must be considered [3,6]. Some potential problems might be represented by higher initial costs, space requirements, infrastructure constraints, increased generation of noise and heat, and the acceptance of the system by the staff [4]. The latter could be particularly critical because during the laboratory automation (LA) for culture-based bacteriology installation, the majority of the microbiological pathways have to be revised and reconsidered according to the new performance offered by the system [7]. Moreover, during the installation of LA, many evaluation trials are required to ensure the adaptation and improvement of old culture protocols to the new system as well as to benefit the most from LA. This latter aspect should be appreciated by microbiologists and perceived as an occasion to rethink microbiology and eliminate some longstanding unnecessary and settled habits in a controlled, reflective, and evidence-based manner [7]. In 2009, we introduced liquid-based microbiology (LBM) to improve the recovery of microorganisms and simplify and reduce the number of microbiological collection devices [2]. LBM allowed us to introduce a Walk Away Specimen Processor (WASP^®^ system, Copan), both with the aim of improving laboratory workflows and to free up human resources that were reallocated to specialized sections of the laboratory (e.g., molecular biological activities) [2]. The COVID-19 pandemic impacted laboratories around the world, and not only because of the need for COVID-19 testing; this proved true in our laboratory as well. The pandemic has in fact heavily increased testing for acute and critical care patients, in particular blood culture sets, respiratory specimens, and urine for culture, as well as the number of susceptibility testing assays required [8,9,10]. The workload had become so difficult that the unique response was to promote automation, particularly if it was impossible to secure additional human resources [6]. In this scenario, LA can be a game-changer [11]. Here, we present our experience with WASPLab^®^ automation (Copan, Brescia, Italy), describing the workflows established and the results obtained one year after LA introduction. In particular, we would like to emphasize how the introduction of LA in the midst of the COVID-19 pandemic has allowed us to respond to diagnostic needs and even to obtain significant improvements in performance, analyzing, in particular, the trends in TAT for blood cultures (BCs) and biological fluids (BLs) before, during, and after LA introduction in our laboratory.

## 2. Materials and Methods

### 2.1. Setting

The setting is a tertiary teaching hospital of the University of Rome “Tor Vergata”, named the Polyclinic of Tor Vergata (PTV). The hospital has 510 beds. The microbiology laboratories at PTV perform approximately 400,000 exams per year. Of these, approximately 200,000 are bacteriological exams, including 36,000 BCs. The microbiology service was 7 days a week, 12 h/day, from 8:00 am to 8:00 pm, in pre-COVID times, but became a 24 h/7-day (24/7) diagnostic service to respond to the emergency caused by the SARS-CoV-2 pandemic. Unfortunately, we did not have sufficient staff to guarantee working on microbiological samples (excluding SARS-CoV-2 specimens, cerebrospinal fluid specimens, and rapid testing for malaria) during the night shift. Therefore, when a blood culture (BC) turns positive during the night, it likely has to wait until the morning to be cultured on the WASPLab^®^ system.

### 2.2. Protocol for WASPLab^®^

The WASPLab^®^ system is a flexible automated specimen processing and reading solution (imaging). The system can be integrated with optional modules, including the PhenoMATRIX™ embedded Artificial Intelligence (AI) and Colibrí™ colony-picking modules. In our laboratory, WASPLab^®^ included AI but not the colony-picking system. WASPLab^®^ installation began in September 2020. The pre-analytical phase for our samples has not undergone any changes subsequent to the introduction of the WASPLab^®^, and we programmed the system so that it could process all biological samples intended for culturing. Specimens delivered to our lab were all collected in LBM devices, namely, ESwab samples, fecal swabs, LIM broth, and SL solution for respiratory specimens (Copan, Brescia, Italy), and in these devices, they are ready to be analyzed in WASPLab^®^. In contrast, broth from positive blood cultures as well as any other fluid specimens, excluding respiratory specimens, are transferred to a Copan instrumentation device (red-cupped sterile tube) and then analyzed by the system. All Copan devices have been used according to the manufacturer’s instructions. Specimens are seeded by the WASP system (a component of the WASPLab^®^ system). WASP has been programmed to seed specimens on different sets of solid media (depending on the nature of the specimen) and with a different type of streaking. Moreover, particularly for respiratory specimens, BCs, fluid specimens from sterile body sites, and genitourinary specimens, the WASP system also prepares smears for Gram staining. The introduction of automation has required a period of revision of microbiological procedures and pathways in use in our laboratories before the WASPLab^®^ installation. Indeed, in the previous three months of WASPLab^®^ implementation (from September 2020 to November 2020), comparative tests were carried out, which allowed us to eliminate redundancies in the seeding of biological samples (e.g., the double plate of blood agar was eliminated and maintained as the only one destinated to be incubated in CO_2_, and pre-enrichment in liquid medium (such as brain heart infusion broth) has been eliminated, e.g., for wound swabs). The final paths, along with the type of seedings, culture media used, and incubation times for the related image acquisitions, are indicated in Appendix A available as an additional file (Grammar WL 124). This includes the type of culture media used for each sample, the type of loop and seeding selected, the expected incubation times and conditions (aerobic or in CO2 enrichment), and finally the timing for digital imaging acquisition. WASPLab^®^ works with incubation times measured in days and hours. Plates are inoculated on day 0 at the hour “XX:XX” and read following the defined timeframe until the end of incubation (determined by the users). The different sets of plates used to seed each sample were defined according to a previous conventional culture procedure and adapted and optimized on the WASPLab^®^ system.

### 2.3. TAT Evaluation

To evaluate the impact of the WASPLab^®^ system on TAT, we divided the samples processed by the system into two macrogroups: BCs and BLs (with the exception of urine samples). TAT was compared for BCs recorded in 2019 (pre-WASPLab^®^ installation), 2020 (during the WASPLab^®^ installation), and 2021 (when WASPLab^®^ was definitely operative). TAT for a BC is defined as the delta time recorded from the time when it was introduced in the continuous monitoring blood culture system (Virtuo system from bioMerieux, Las Balmas, France) to that of the final report available for the clinicians. However, as mentioned before, the lack of sufficient staffing prevented us from guaranteeing steady monitoring of the BCs during night shifts. Given this uncertainty, we decided to evaluate the TAT from BC (2019–2021) only among the 12 h/day shifts, avoiding altogether any possible bias. TAT for cultures (such as respiratory samples, fluid specimens from body sites, swabs, etc., but not urine) recorded in 2019–2021 was compared as described for BCs and similarly refers to the time lapse from the check-in to the lab to the time of final reporting to the clinicians. The time to positivity (TP) for a BC was calculated using Myla software (v. 4.0) (bioMérieux), with which the Virtuo system is equipped. Similarly, TAT for BLs was evaluated by comparing those recorded in 2019, 2020, and 2021. For a BL, TAT is the delta time recorded from the time when the check-in occurred in the lab to that of the final report available for the clinicians.

### 2.4. Lab Professionals Involved in the Project

The installation of the system began at the end of September 2020 and concluded in November 2020. The implementation of the system was divided into several phases: the first was the installation (by the Copan staff), which took approximately two weeks; the second was the programming and consolidation of the culture protocol, through the initial definition and adaptation of the seeding protocols and reading protocols; and finally, validation of the protocols. During phase two, a microbiologist among the graduated staff dedicated himself to the realization of the project alongside the supplier’s specialist. Phase three entailed verification that all the protocols tested and validated in the previous two phases were fully operational for the entire range of possible biological samples received during routine and urgent activity by the laboratory. During the third phase, four additional experienced technicians entered the training group, denoted “skiller”, in addition to the microbiologist already employed in phase two. In phase 4, i.e., the total operation of the WASPLab^®^ system, all the staff were trained and involved. In phase 4, the specialist from the supplier company remained available 12 h/day until the end of the installation to resolve any critical issues and integrate training whenever required in the field.

### 2.5. Statistical Evaluation

Data recorded for each BC as well as for other BLs processed were extracted by the laboratory LIS (Instrumentation Laboratory S.p.A.—Werfen, Viale Monza, Italy) and then evaluated considering the median, harmonic mean, average, and standard deviation. To evaluate whether the WASPLab^®^ system impacted TAT, we evaluated and compared the TAT recorded in three different periods: 2019, 2020, and 2021, as described above. The comparison of the recorded TAT was performed using a *z*-test. Differences were considered significant for *p* values ≤ 0.05.

### 2.6. Pathogen Identification, Antimicrobial Susceptibility Testing, and Molecular Assays

Microorganisms obtained in our cultures were all identified using a MALDI TOF MS System (Bruker Daltonics, Bremen, Germany), while antimicrobial susceptibility testing (AST) was performed using Micronaut panels (Diagnostika Gmbh, Bornheim, Germany, now a subsidiary of Bruker Daltonics, Billerica, MA, USA) run on MICRO MIB (Bruker Daltonics, Billerica, MA, USA). For some BCs, particularly those from critically ill patients, a molecular assay was performed to produce a result quickly available to the clinician. The BCID syndromic panel was used, namely, ePlex^®^ Panels (GenMark Diagnostics, Inc., Carlsbad, CA, USA). The panels provide pathogen identification and the presence/absence of the main resistance genes after approximately 1 h and 15 min.

## 3. Results

Table 1 and Table 2 report the TAT values observed for BCs and cultures BLs, respectively.

These tables also report the total number of positive samples examined in each year. TAT values are expressed in days and hours. The median, harmonic mean, and average TAT show overall declines from 2019 (pre-WASPLab^®^ installation) to 2021 (when WASPLab^®^ became fully operative) for both BCs and BLs.

In Figure 1, the global trend in TAT for BCs and BLs from 2019 to 2021 is reported. In the case of the TAT trend for BCs, as shown in the figure, the median and harmonic mean overlap.

The TAT trend graphics from BCs and from BLs, showing the data’s dispersion of each point, are reported in Figure 2a,b, respectively. In Figure 2a, the black arrow shows the tendency line for BCs collected in 2021 whereas, in Figure 2b, the black arrow shows the tendency line for cultures collected in 2021.

The TAT averages obtained for BCs decreased over 2019 to 2021 from 97 h to 53.5 h, with a savings of approximately 43.5 h in the total workflow for this sample type. Similarly, for BLs, the changes over the years in TAT were significant since the mean TAT was 73 h before the introduction of WASPLab^®^ and 58 h at full system operation (Δ20 h). The decrease in TAT was statistically significant, with a *p*-value of 0.03 and 0.008 for BCs and BLs, respectively. As reported above, TAT was evaluated as the time frame from the delivery of a sample to the lab to the final report, but for BCs, we also considered the time to positivity (TP) recorded for each BC. The latter is the time between the introduction of a BC into the monitoring incubation system (that is, in our instance, the Virtuo system) and the time when the same sample turned positive. For our BCs, TPs ranged from a minimum value of 1 h and 44 min to a maximum of 119 h and 4 min (average 17 h and 42 min), and no significant differences were observed during the observational time of our study (2019–2021). This implies that the average time required to process a positive BC is approximately 36.5 h, which is the difference between the averages of BC TAT (53.5 h) and TP (17 h). This 36.5 h also included the time required for pathogen culturing, microbial identification, and antimicrobial susceptibility testing. The continuous incubation system allows for more efficient microbial growth, so that pathogen ID/AST can therefore be obtained much earlier as compared to the traditional work-up, potentially improving the turnaround time. Figure 3 shows an example of the workflow for a positive blood culture under the condition that the staff is free to operate also during the night shift; the TAT was reduced to 33 h and 42 min (this includes the time to positivity).

Another aspect to evaluate in LA is the set of advantages introduced by digital imaging. Figure 4 presents an example of mixed culture that the digital imaging of the system allowed us to appreciate. The image magnification was obtained using the zoom of the WASPLab^®^ by Copan imaging system.

## 4. Discussion

The COVID-19 pandemic has strained microbiology laboratories already overburdened with important workloads and insufficiently staffed [5,12].

The result obtained due to the WASPLab^®^ introduction in our lab is particularly relevant because it occurred during the COVID-19 pandemic, when the greatest allotment of human resources was employed and dedicated to processing nasopharyngeal swabs for SAR-CoV-2 detection.

Before embarking on the path of total automation, it is wise to carefully evaluate workflows and diagnostic paths to understand what type of automation can be of benefit to each laboratory structure and what degree of automation should be pursued [5,13,14]. With continuous automated incubation of plated media, a laboratory can achieve a smoother workflow, albeit usually not including anaerobic or microaerophilic conditions. The implementation of total lab automation entails auditing and reviewing of longstanding routines. Some assays that are easy to perform in a conventional system are cumbersome and inconvenient to perform within an automated workflow [15]. Moreover, it is well known that automation is effective in increasing productivity, reducing manual hands-on time per sample, decreasing turnaround time, improving operator safety from exposure to hazardous materials, minimizing errors, and enhancing patient safety, as well as reducing costs and the need for specific types of training [4]. On the other hand, for truly effective LA, it is very important to associate automation with a 24/7 working regime because this is the only way to achieve the most out of the technology [16]. A significant reduction in TAT has already been discussed and presented by Cherkaoui et al., but to our knowledge, this is the first report in which TAT was examined for both BCs and fluid sample cultures [17].

We can speculate that our results, the reductions in TAT of approximately 43.5 and 20 h, respectively, for BC and BL samples, should be construed as an even more striking result, being obtained under the worst working conditions ever. Despite the introduction of the WASPLab^®^ system, we have not been able to reduce the number of technical personnel units dedicated to the microbiology lab, probably because the laboratory activity was intense and tiring due to the surplus of activities caused by COVID-19. In contrast, the automation allowed us to be able to allocate newly introduced technical personnel, not fully trained in the WASPLab^®^, who, given the simplicity of the system, were able to perform the activity of bacteriology with the support, whenever needed, of a single skilled technician. It is important to underline that the results obtained for BCs are very relevant, especially if it is considered that the statistical evaluation was performed excluding the BCs processed during the night shift. Apart from the TAT reduction, visualizing everyday cultures thanks to digital imaging was truly an advantage for microbiologists and clinicians, who could receive more detailed reports in less time [18].

However, since the described TATs are purely laboratory ones, it was not possible to evaluate the impact of automation on clinical TATs. Given that our hospital does not have a fully operational computerized medical record, we were not fully able to make an accurate assessment of the impact of the anticipation of our results on clinical choices. We can only report anecdotal findings that, in the face of a timely result, there was some correspondence in the response to the clinician, who, observing the result on the LIS, turned to the laboratory to receive further information on the progress and/or request further information on the drug susceptibility.

Future studies are required to record a cost-effective evaluation of LA by also evaluating the outcome.

To conclude, automation, alongside multiple benefits (improvements in standardization, reproducibility, and laboratory TAT reduction), is accompanied by limitations, including obsolescence, the cost of initial investment, and the fact that large high-tech robotic solutions may not be appropriate for medium- or low-dimension-laboratories with constrained financial and spatial resources. For the latter case, the centralization of samples from peripheral laboratories to large laboratories that have access to this type of technology is conceivable. This is already a reality in Italy (with the so-called large-area laboratories that collect samples from small laboratories close to them). In these cases, however, an efficient sample transport system must be designed and structured which, in any case, can have a negative impact on the laboratory TATs of “peripheral” samples.

## 5. Conclusions

The WASPlab system requires minimal weekly ordinary maintenance; it is just a matter of keeping clean the conveyors, the sensor system, as well as the plate housing where the photographs are taken. Therefore, the commitment of the staff is of little significance. Breakdowns are very rare, and when they do occur, they are mostly resolved with remote assistance (available 12 h a day on Saturdays and holidays until 6 pm). Only in exceptional cases (an average of one/two years depending on the workload on the system) are never blocking; remote assistance puts the laboratory in a position to continue the activity pending on-site intervention, which is guaranteed by the next morning.

WASPLab^®^ not only allowed us to optimize our workflow but also allowed potentially life-saving switches in antibiotic regimens to be initiated sooner [18,19]. In fact, our example of the BC workflow during the night shift, which we have reported, clearly shows that we can reduce TAT on BC by 33 h, saving approximately 20 h with respect to the average TAT of 53 h, calculated after WASPLab’s introduction, or that of 95 h, recorded before LA. This precious time can be dedicated to the time necessary to set up an antibiogram in broth microdilution according to the required operating standards [19,20].

Furthermore, if we imagine pairing the fast track due to LA with the rapid antimicrobial susceptibility test (RAST), we can hypothesize that the microbiologist can be truly effective and have an impact on the medical decision only a few hours after receiving a sample in the laboratory [21,22]. However, LA in our lab remains an ongoing experience, not a tale; in fact, we are now preparing artificial intelligence (PhenoMATRIX™) to make decisions on our behalf after having recorded our habits in processing and making decisions on the work-up of our specimens. This is the most rewarding part of our path because it will allow us to further streamline the steps that are still closely linked to the presence of the operator, and we imagine being able to further reduce TAT when the system is at full capacity [11].

## Figures and Tables

**Figure 1 diagnostics-13-02243-f001:**
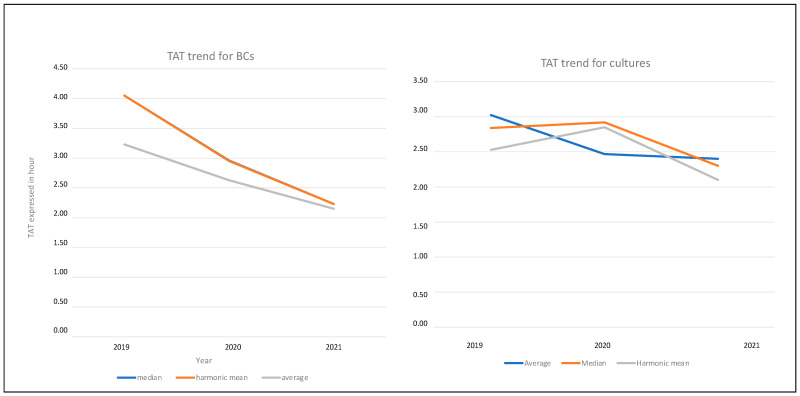
TAT trend for BCs and for BLs.

**Figure 2 diagnostics-13-02243-f002:**
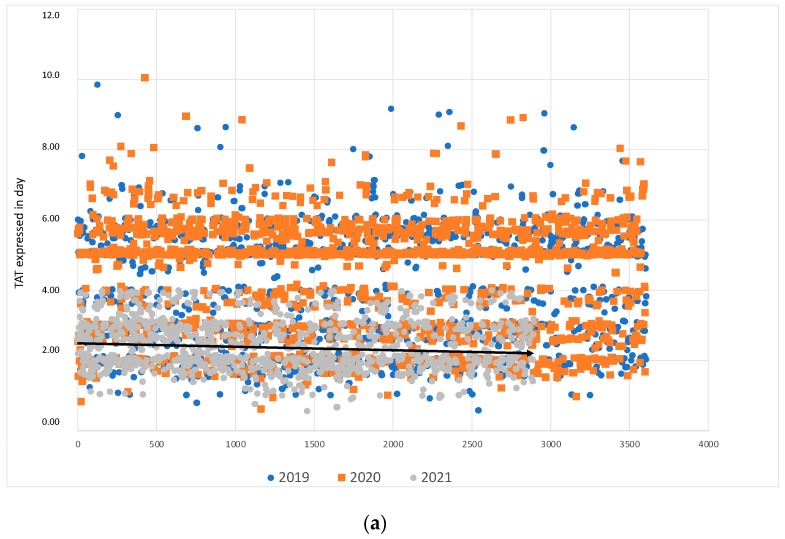
(**a**) The TAT trend graphics from BCs showing the data’s dispersion at each point. (**b**) The TAT trend graphics from BLs showing the data’s dispersion at each point.

**Figure 3 diagnostics-13-02243-f003:**
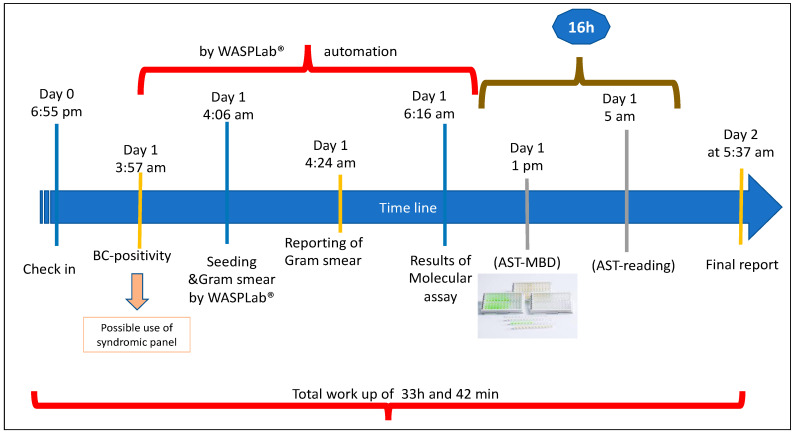
Workflow on blood culture during night shift.

**Figure 4 diagnostics-13-02243-f004:**
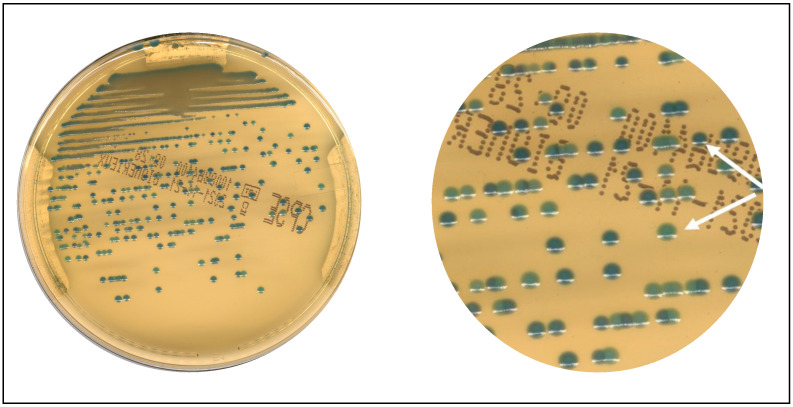
Imaging—enhanced reality. Box (**left**): an apparently pure culture; Box (**right**): a magnification of the image of the same culture showing two different colonies indicated by the arrows.

**Table 1 diagnostics-13-02243-t001:** TAT trend recorded in 2019–2021 for BCs.

Statistical Evaluation	2019		2020		2021	
	TAT * 2019	TAT § 2019	TAT * 2020	TAT § 2020	TAT * 2021	TAT § 2021
Average	4.05	97	2.96	71.04	2.23	53.52
Median	4.05	97	2.95	70.80	2.23	53.52
Harmonic mean	3.23	77	2.63	63.12	2.15	51.60
Standard Deviation	1.56	37	1.39	33.36	1.09	26.16
No. of positive BCs	3613		2934		1732	

* TAT from the arrival in the lab to the final report (available for the clinicians) expressed in days; § TAT expressed in hours.

**Table 2 diagnostics-13-02243-t002:** TAT trend from BLs recorded in 2019–2021.

Statistical Evaluation	2019		2020		2021	
	TAT * 2019	TAT § 2019	TAT * 2020	TAT § 2020	TAT * 2021	TAT § 2021
Average	3.02	73	2.47	59	2.40	58
Median	2.84	68	2.92	70	2.30	55
Harmonic mean	2.53	61	2.85	68	2.10	50
Standard Deviation	1.36	33	0.45	11	1.00	24
No. of positive cultures	2299		1140		2768	

* TAT from the arrival in the lab to the final report (available for the clinicians) expressed in days; § TAT expressed in hours.

## Data Availability

Data availability statements are available in the article.

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
