# Peer review of "Laboratory Automation in Microbiology: Impact on Turnaround Time of Microbiological Samples in COVID Time"

_diagnostics, 2023, doi:10.3390/diagnostics13132243_

Round 1
Reviewer 1 Report
- Workflow and methods used in the lab are described very roughly, a process map indicating which steps are manual and which steps are automated by the WASPlab system would be very helpful for the reader to understand the impact on TAT and staff hours
- the authors describe a decrease in TAT by roughly 45 % for BCs and 20% for LBs, however 26% of this decrease for BCs is already achieved in 2020 even though the WASPLab was only fully operational by Nov2020. It seems, the improvement of TAT is as much due to other factors as due to laboratory automation. The authors claim a significant reduction in TAT due to laboratory automation, however a large part of this reduction was carried out in a year, where the automated system was only fully operational for one month. This is insufficiently discussed in the text.
- the authors mention in the introduction, that during the Covid19 pandemic workload in microbiology labs increased. however, the data in table 1 and 2 shows that the number of positive specimens decreased from 2019 to 2021. The authors need to elaborate on the discussion of this effect.
- the Figures seem to be compressed and not imported correctly into the document. In Figure 1 an axis description for the y axis and x axis is missing and the two parts of the figure stretch across two pages, and it is unclear what "TAT trend for BCs" and "TAT trend for cultures" is in relation to the text. The median line is not visible at all, it seems to be hidden behind the harmonic mean, more formatting is required for this figure.
- a two tailed Fishers exact test was used to determine whether the decrease of TAT is significant. It is not mentioned which years are compared for this test (2019 with 2020 or 2021). The authors should elaborate why this statistical method in particular is used.
- In Figure 2 the workflow of a blood culture is shown, it is unclear which part has been automated by the WASPlab technology and how staff is freed during this period by laboratory automation
- In Figure 3 an imaging system is showing two different colonies that have been identified by the enhanced reality. The authors need to elaborate on the method this image was evaluated with, whether software or staff was used to draw this conclusion from the image. If software was used, the authors should show validation data to underline whether this technology is reproducible.
- increased use of filler words
- unnecessarily complex sentence structure
- the paragraphs from line 349 - 355 and 356 - 361 seem are redundant
Author Response
-

Reviewer 2 Report
The paper deals with an important issue for the microbiology laboratory.
I would suggest some refinements mainly in the description of the methods and in the presentation of the results. First, the work would be easier to understand by providing more details on the differences in the workflow before and after the introduction of automation.
Second, in the trend graphics the dispersion of data of each point should be added to show the variability of results. Accordingly, drawbacks if any of the automated system should be discussed, e.g. robustness of the instrument, maintenance needed, time losses during the process, etc
Third, in the discussion possible advantages of both pre-analytical and post-analytical phase automation should be outlined, e.g. number of unprocessable samples, rate of uncertain results or expert system validation.
Minor point: the last paragraph is repeated twice.
Author Response
-

Reviewer 3 Report
Total laboratory automation in clinical microbiology is expected to reduce turn-around times, and this is the case in this manuscript.
This is an empirical data observation in the real world of an European hospital of five hundred patients.
Major remarks:
Limits of your approach should be better put in evidence peculiarly in the Discussion:
a.- Laboratory TAT as used is your manuscript is not equivalent to Clinical TAT, this last one being more relevant for the patient; this should be clearly stated as limit to your analysis;
b.- Which conclusion for small volume labs? In rural parts of the country? The productivity of big labs will increase, but there is a risk for closing small rural (peripheral) labs, increasing the medical desertification in rural population - and increasing clinical TAT if samples have to be transported from rural villages to Roma. Such a risk shoud be clearly identified;
c.- Dependence of big pharma: are these "new" automation developments adapted to recognize new and difficult pathogens? (Example: Candida auris). There is a risk of losing expertise with capacity of detecting "new" pathogens.
d.- Why do you use "harmonic means"? If it is statistically useful, describe the reason for its use in the "Methods" section. If not useful, delete it.
Suggestions:
a) Discard opinion words such as "...revolutionizing 28 laboratory workflows..." (lines 28-29); "...Microbiologists are unfortunately accustomed to working under conditions of scarce human resources, and for this reason, promoting automation processes is important" (lines 285-287) and other sentences with emotional charge, but without interest for the reader.
Discussion can be abridged and the limits described above can be more clearly discussed. (Look at Word File added)
Conclusion is not useful in this case.
Editing:
1) Lines 356-361 duplicate lines 349-354;
2) Many abbreviations: at least in the legends of the tables and figures, the total wording "Turn-around time" should be used for convenience of the readers.

Author Response
-

Round 2
Reviewer 1 Report
The authors have addressed the comments of the last review. I suggest to accept the manuscript in its current form.